# Continuous Subspace Optimization
# for Continual Learning

**Quan Cheng**[1,2], **Yuanyu Wan**[3,4], **Lingyu Wu**[1,2], **Chenping Hou**[5], **Lijun Zhang**[1,2,*]

[1]National Key Laboratory for Novel Software Technology, Nanjing University, Nanjing, China
[2]School of Artificial Intelligence, Nanjing University, Nanjing, China
[3]School of Software Technology, Zhejiang University, Ningbo, China
[4]Hangzhou High-Tech Zone (Binjiang) Institute of Blockchain and Data Security, Hangzhou, China
[5]College of Science, National University of Defense Technology, Changsha, China
{chengq, wuly, zhanglj}@lamda.nju.edu.cn
wanyy@zju.edu.cn, hcpnudt@hotmail.com

## Abstract

Continual learning aims to learn multiple tasks sequentially while preserving prior knowledge, but faces the challenge of catastrophic forgetting when adapting to new tasks. Recently, approaches leveraging pre-trained models have gained increasing popularity in mitigating this issue, due to the strong generalization ability of foundation models. To adjust pre-trained models for new tasks, existing methods usually employ low-rank adaptation, which restricts parameter updates to a fixed low-rank subspace. However, constraining the optimization space inherently compromises the model's learning capacity, resulting in inferior performance. To address this limitation, we propose **Co**ntinuous **S**ubspace **O**ptimization for Continual Learning (CoSO) to fine-tune the model in a series of subspaces rather than a single one. These sequential subspaces are dynamically determined through the singular value decomposition of the gradients. CoSO updates the model by projecting gradients onto these subspaces, ensuring memory-efficient optimization. To mitigate forgetting, the optimization subspace of each task is constrained to be orthogonal to the historical task subspace. During task learning, CoSO maintains a task-specific component that captures the critical update directions for the current task. Upon completing a task, this component is used to update the historical task subspace, laying the groundwork for subsequent learning. Extensive experiments on multiple datasets demonstrate that CoSO significantly outperforms state-of-the-art methods, especially in challenging scenarios with long task sequences.

## 1 Introduction

Deep neural networks have achieved remarkable success when trained on large-scale offline data under the assumption of independent and identically distributed (i.i.d.) samples [He et al., 2016, Vaswani et al., 2017, Dosovitskiy et al., 2021]. However, real-world applications often require models to learn from a sequence of tasks with different data distributions, a scenario known as continual learning [De Lange et al., 2022, Van de Ven et al., 2022, Masana et al., 2022, Wang et al., 2024, Zhou et al., 2024b]. The major challenge in continual learning is catastrophic forgetting [McCloskey and Cohen, 1989], where the model's performance on previously learned tasks deteriorates significantly as it adapts to new tasks.

In recent years, pre-trained models especially vision transformers (ViTs) [Dosovitskiy et al., 2021] have demonstrated exceptional performance across various downstream tasks through their robust

---

*Lijun Zhang is the corresponding author.

generalization ability. This property makes pre-trained models highly promising in mitigating catastrophic forgetting, leading to a growing research focus on continual learning with foundation models [Smith et al., 2023, Lu et al., 2024, Liang and Li, 2024, Zhou et al., 2024a, Wu et al., 2025]. To efficiently fine-tune pre-trained ViTs, existing continual learning methods [Gao et al., 2023, Liang and Li, 2024, Wu et al., 2025] employ low-rank adaptation (LoRA) [Hu et al., 2022] to optimize the models, which confine parameter updates to a specific low-rank subspace to reduce the interference between tasks. However, this rigid constraint on update directions inherently limits the model's learning capacity, leading to inferior performance.

To address this issue, we propose **Co**ntinuous **S**ubspace **O**ptimization for Continual Learning (CoSO), which achieves enhanced adaptability by optimizing the model within multiple subspaces rather than a fixed one. These sequential subspaces are derived from the singular value decomposition of the gradients. By projecting gradients onto these low-dimensional subspaces for Adam [Kingma, 2014] optimization and then projecting back for parameter updates, CoSO achieves memory-efficient learning. To prevent forgetting, we enforce orthogonality between the optimization subspaces of current and historical tasks during training. While learning a task, CoSO leverages Frequent Directions (FD) [Ghashami et al., 2016, Wan and Zhang, 2018, 2022] to maintain a compact task-specific component, which captures critical update directions of the current task with negligible computational cost. After completing the current task, we use this dedicated component to estimate the task-specific subspace, which is then integrated into the historical task subspace, laying the groundwork for subsequent learning.

Experimental results on CIFAR100, ImageNet-R, and DomainNet show that CoSO consistently outperforms state-of-the-art methods by a significant margin across diverse continual learning settings, particularly in challenging scenarios involving long task sequences. The substantial performance gains highlight CoSO's strong potential for real-world continual learning.

In summary, our contributions are as follows:

- We propose CoSO, a novel continual learning framework that fine-tunes pre-trained models via continuous gradient-derived subspaces, enabling efficient adaptation to sequential tasks.
- We introduce a lightweight mechanism to maintain the historical task subspace, enabling CoSO to keep current updates orthogonal to the historical subspace and thereby mitigate task interference.
- We conduct extensive experiments, demonstrating CoSO's superior performance over prior PEFT-based continual learning methods across various datasets and settings.

## 2    Related Work

In this section, we review related work on continual learning and low-rank optimization in offline learning.

### 2.1    Continual Learning

Continual learning [De Lange et al., 2022, Van de Ven et al., 2022, Masana et al., 2022, Wang et al., 2024, Zhou et al., 2024b] aims to enable neural networks to incrementally learn from a sequence of tasks while retaining previously learned knowledge. These approaches broadly fall into five categories [Wang et al., 2024]: regularization-based methods [Zenke et al., 2017, Kirkpatrick et al., 2017, Li and Hoiem, 2017], replay-based methods [Lopez-Paz and Ranzato, 2017, Rebuffi et al., 2017, Chaudhry et al., 2019a,b, Liu et al., 2020, Sun et al., 2022], optimization-based methods [Farajtabar et al., 2020, Saha et al., 2021, Wang et al., 2021], representation-based methods [Madaan et al., 2022, Pham et al., 2024], and architecture-based methods [Yoon et al., 2018, Li et al., 2019, Sokar et al., 2021, Liang and Li, 2023]. Regularization-based methods introduce additional loss terms to constrain parameter updates, preventing drastic changes in parameters that are important for early tasks. Replay-based methods store a small subset of training samples from previous tasks in a limited buffer and periodically replay these samples alongside new data, allowing the model to rehearse earlier knowledge. Optimization-based methods manipulate the update directions of each task according to preserved information of previous tasks. Representation-based methods utilize statistical information of features to calibrate classifiers. Architecture-based methods dynamically modify network architectures, dedicating specific model capacity for new tasks.

Early continual learning approaches typically initialize their models with random weights. The strong generalization capabilities of foundation models, especially vision transformers [Dosovitskiy et al., 2021], have made pre-trained architectures an increasingly attractive solution for continual learning [Zhou et al., 2024a]. Recent developments in parameter-efficient fine-tuning (PEFT) based continual learning methods [Gao et al., 2023, Liang and Li, 2024, Lu et al., 2024, Wu et al., 2025] have facilitated efficient adaptation of foundation models through selective parameter optimization, substantially lowering computational requirements. Existing PEFT-based methods can be broadly categorized into two groups: (1) prompt-based techniques that focus on optimizing learnable tokens [Lester et al., 2021, Wang et al., 2022a,b, Smith et al., 2023, Lu et al., 2024], and (2) LoRA-based methods that adjust parameters within constrained low-dimensional subspaces [Gao et al., 2023, Liang and Li, 2024, Wu et al., 2025].

Among prompt-based approaches, L2P [Wang et al., 2022b] introduces task-specific prompt tokens to modulate the pre-trained model's behavior, but struggles with knowledge transfer between tasks. DualPrompt [Wang et al., 2022a] addresses this limitation by maintaining both task-specific and task-invariant prompts, enabling better knowledge sharing. CODA-Prompt [Smith et al., 2023] further enhances adaptation flexibility through dynamic prompt composition from a shared pool. VPT-NSP$^2$ [Lu et al., 2024] learns each task by tuning learnable prompts in the null space of previous tasks' features. However, these methods influence model behavior indirectly through learnable tokens, which restrict the model's ability to capture complex task-specific features.

Complementary to prompt-based methods, LoRA-based approaches directly update model parameters in a parameter-efficient manner. InfLoRA [Liang and Li, 2024] constrains the parameter updates within a predetermined subspace to reduce the interference between tasks. SD-LoRA [Wu et al., 2025] decouples the learning of the magnitude and direction of LoRA components. However, both methods confine weight updates to a specific low-rank subspace, which inherently limits the model's learning capacity. Unlike these methods, CoSO updates the parameters across a series of subspaces, enabling the learning of full-rank weights and thereby enhancing the model's flexibility.

## 2.2 Low-rank Optimization in Offline Learning

Low-rank adaptation (LoRA) [Hu et al., 2022] has gained significant attention for its ability to reduce computational and memory requirements when fine-tuning pre-trained models [Mao et al., 2022, Zhang et al., 2023]. Specifically, LoRA reparameterizes the update of a linear layer's weights $\Delta W = BA \in \mathbb{R}^{m \times n}$, where $B \in \mathbb{R}^{m \times r}, A \in \mathbb{R}^{r \times n}$ are low-rank matrices. By freezing the original weights and only updating the low-rank components, LoRA enables parameter-efficient fine-tuning while preserving performance in many downstream tasks. However, it has been demonstrated [Xia et al., 2024] that low-rank weight updates limit the performance compared to full-rank fine-tuning. Recent works [Cosson et al., 2023, Zhao et al., 2024] have shown that neural network gradients often exhibit low-rank structure. Instead of approximating the weight matrix as low rank, GaLore [Zhao et al., 2024] directly leverages the low-rank gradients to optimize the model. This methodology enables memory-efficient optimization through effective dimensionality reduction in gradient spaces.

To be concrete, GaLore utilizes the singular value decomposition (SVD) of $G_t \in \mathbb{R}^{m \times n}$ to compute a low-rank projection matrix $P_t \in \mathbb{R}^{m \times r}$, where $r \ll n$ is the target rank. Leveraging $P_t$, GaLore transforms the gradient $G_t$ into a compact form $P_t^\top G_t$ to achieve memory-efficient parameter updates. At each training step $t$, the gradient update can be decomposed into three operations:

$$\begin{aligned} R_t &= P_t^\top G_t && \text{(forward projection)} \\ N_t &= \text{Adam}(R_t) && \text{(adam optimizer update)} \\ \tilde{G}_t &= P_t N_t. && \text{(backward projection)} \end{aligned}$$

The projection matrix $P_t$ is periodically updated through SVD to follow the evolving gradient subspace. Utilizing the final gradient $\tilde{G}_t$, GaLore updates the model parameters with learning rate $\eta$:

$$W_t = W_{t-1} - \eta \tilde{G}_t.$$

Compared to LoRA, GaLore not only reduces memory storage from $(mn + 3mr + 3nr)$ to $(mn + mr + 2nr)$, but also achieves higher model capacity by directly optimizing in the most relevant gradient subspaces rather than constraining updates to a predefined low-rank structure.

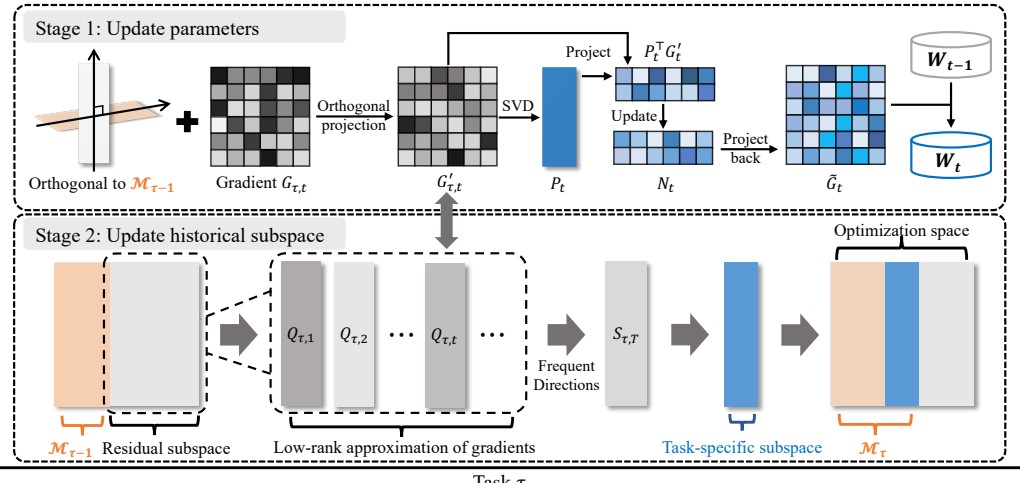

Figure 1: CoSO optimizes the parameters in continual low-rank subspaces, enhancing the learning capacity of models. To mitigate forgetting, the optimization subspaces of the current task are set to be orthogonal to the historical task subspace. While learning a task, CoSO consolidates the low-rank approximation matrices $\{Q_{\tau,t}\}_{t=1}^{T}$ into a task-specific component $S_{\tau,T}$ through Frequent Directions. The dedicated component is then used to update the historical task subspace spanned by $\mathcal{M}_{\tau-1}$.

## 3 Methodology

In this section, we first introduce the necessary preliminaries, then present the details of our approach.

### 3.1 Preliminaries

In continual learning, a model needs to learn a sequence of tasks while retaining knowledge of previous tasks. We consider the class-incremental learning setting, where task identities are unavailable at inference time and access to historical data is prohibited during learning new tasks [Wang et al., 2022a, Smith et al., 2023, Liang and Li, 2024, Lu et al., 2024, Wu et al., 2025]. We denote the task sequence as $\mathcal{D} = \{\mathcal{D}_1, ..., \mathcal{D}_N\}$, where each task dataset $\mathcal{D}_\tau = \{(\mathbf{x}_{i,\tau}, y_{i,\tau})\}_{i=1}^{n_\tau}$ contains $n_\tau$ input-label pairs. Following recent work [Wang et al., 2022a, Gao et al., 2023], we adopt a pre-trained Vision Transformer (ViT) [Dosovitskiy et al., 2021] as the backbone network, denoted as $f_\Theta(\cdot)$ with parameters $\Theta$, and classifier $h_\Phi(\cdot)$ with parameters $\Phi$, thus the model is $h_\Phi(f_\Theta(\cdot))$. Formally, the hidden state $Y_\tau^\ell$ of feature $X_\tau^\ell$ at the linear layer $\ell$, can be calculated as $Y_\tau^\ell = W^\ell X_\tau^\ell$, where $W^\ell$ is the weight matrix of the linear layer. Let $G_{\tau,t}^\ell$ denote the gradient at the $t$-th training step of the linear layer $\ell$ in task $\tau$. For simplicity, we omit the symbol $\ell$, using $W$ to refer $W^\ell$ and $G_{\tau,t}$ to refer $G_{\tau,t}^\ell$ in the following sections.

### 3.2 Continuous Subspaces Optimization

Inspired by GaLore [Zhao et al., 2024], we propose CoSO to address the rigidity of single subspace adaptation methods through multiple subspaces optimizing. However, directly using GaLore causes severe interference between different tasks in continual learning. To minimize the interference, CoSO enforces orthogonal constraints between current and historical subspace during training. Motivated by memory consolidation in cognitive neuroscience [Dudai, 2004], CoSO estimates a task-specific subspace to consolidate knowledge upon learning each task. This subspace preserves critical learning directions of the task based on gradients at all training steps, and is incrementally integrated into the historical task subspace, enabling efficient knowledge accumulation. The whole process of CoSO is illustrated in Figure 1. We first introduce how to optimize the model in continuous subspaces in this section. Then we present how to update the historical task subspace in Section 3.3.

In continual learning, the key challenge is to prevent new task updates from interfering with previously learned knowledge. Building on the insight that gradient updates in neural networks typically lie in the span of input features [Saha et al., 2021], we develop an approach that leverages this gradient-input feature relationship to minimize task interference through orthogonal projection. Specifically, we maintain an orthogonal basis matrix $\mathcal{M}_{\tau-1}$ that spans the gradient subspace accumulated from all previous tasks prior to the current task $\tau$. Since gradients inherently encode information about the input features they were computed from, this historical subspace captures the principal directions that were important for learning previous tasks. We recognize that gradient steps along these historical directions would cause maximal interference with past learning, while gradient steps orthogonal to this space result in minimal interference. For each gradient $G_{\tau,t}$ computed during training on task $\tau$, we project it onto the orthogonal complement of historical subspace:

$$G'_{\tau,t} = G_{\tau,t} - \mathcal{M}_{\tau-1}\mathcal{M}_{\tau-1}^\top G_{\tau,t}. \tag{1}$$

This projection removes the gradient component aligned with the learning directions of previous tasks, leaving only the orthogonal component $G'_{\tau,t}$ for updating the parameters. When we update the weight matrix using these orthogonal gradients, i.e., $\Delta W = -\eta \sum_{t=1}^T G'_{\tau,t}$, the parameter changes occur in directions that have minimal overlap with the optimization trajectories of previous tasks. This approach effectively partitions the parameter space, preserving directions important for past tasks while utilizing orthogonal directions for new learning. The orthogonal projection thus provides a principled way to balance plasticity for new tasks with stability for old tasks, enabling the model to expand its capabilities while mitigating interference.

However, updating the model with the full orthogonal gradient $G'_{\tau,t}$ incurs substantial memory overhead and high computational cost, particularly in vision transformers. To achieve memory-efficient fine-tuning, we follow GaLore [Zhao et al., 2024] and decompose $G'_{\tau,t} \in \mathbb{R}^{m \times n}$ using singular value decomposition (SVD) to get the projection matrix $P_{\tau,t}$:

$$\begin{aligned} U\Sigma V^\top &= \mathrm{SVD}_{r_1}(G'_{\tau,t}) \\ P_{\tau,t} &= U[:, : r_1], \end{aligned} \tag{2}$$

where $m$ and $n$ are the dimensions of the original weight matrix, $r_1 \ll n$ is the target projection rank, and $\mathrm{SVD}_{r_1}(\cdot)$ denotes a truncated SVD that retains the top-$r_1$ singular values. Subsequently, we project the orthogonalized gradient $G'_{\tau,t}$ into the low-rank subspace spanned by $P_{\tau,t}$, effectively reducing the memory footprint of parameter updates:

$$R_{\tau,t} = P_{\tau,t}^\top G'_{\tau,t}. \tag{3}$$

Then $R_{\tau,t}$ is updated by Adam [Kingma, 2014] as follows:

$$\begin{aligned} M_{\tau,t} &= \left(\beta_1 \cdot M_{\tau,t-1} + (1-\beta_1) \cdot R_{\tau,t}\right)/(1-\beta_1^t) \\ V_{\tau,t} &= \left(\beta_2 \cdot V_{\tau,t-1} + (1-\beta_2) \cdot R_{\tau,t}^2\right)/(1-\beta_2^t) \\ N_{\tau,t} &= M_{\tau,t}/\left(\sqrt{V_{\tau,t}} + \epsilon\right), \end{aligned} \tag{4}$$

where $\beta_1, \beta_2$ are decay rates, $M_{\tau,t}$ is the first-order momentum, and $V_{\tau,t}$ is the second-order momentum. The low-rank normalized gradient $N_{\tau,t}$ is then projected back to update the parameters with learning rate $\eta$:

$$\begin{aligned} \tilde{G}_{\tau,t} &= P_{\tau,t} N_{\tau,t} \\ W_{\tau,t} &= W_{\tau,t-1} - \eta \cdot \tilde{G}_{\tau,t}. \end{aligned} \tag{5}$$

Because $P_{\tau,t}$ is computed from the projected gradient $G'_{\tau,t}$, which is orthogonal to the historical subspace spanned by $\mathcal{M}_{\tau-1}$, any parameter updates derived from $P_{\tau,t}$ remain in the null space of previous tasks' feature spaces. Consequently, the linear layer's output for every earlier task remains unchanged, preventing interference at the representation level. Since $P_{\tau,t}$ is dynamically changed to capture the most important directions of $G'_{\tau,t}$, we are optimizing the model in continuous subspaces rather than a fixed one, thereby expanding the model's representational adaptability. To balance computational efficiency, we update the projection matrix $P_{\tau,t}$ every $K$ steps. By updating $R_{\tau,t}$ in lower dimension space, the memory requirement is reduced from $(mn + 3mr_1 + 3nr_1)$ to $(mn + mr_1 + 2nr_1)$ compared to LoRA-based methods, such as InfLoRA [Liang and Li, 2024] and SD-LoRA [Wu et al., 2025].

### 3.3 Historical Task Space Update

**Task-Specific Subspace Estimation.** To update the orthogonal basis matrix $\mathcal{M}_{\tau-1}$ of the historical task space, we need to efficiently estimate a task-specific subspace, which retains the critical gradient directions of the current task. Specifically, for task $\tau$, the model undergoes $T$ training steps, producing a sequence of gradients $\{G'_{\tau,1}, ..., G'_{\tau,T}\}$, where $G'_{\tau,t} \in \mathbb{R}^{m \times n}$. To identify the primary directions of these gradients, we consider the accumulated covariance matrix $\sum_{t=1}^{T} G'_{\tau,t} G'^{\top}_{\tau,t} \in \mathbb{R}^{m \times m}$, which integrates information from all training steps and characterizes the subspace where most updates occur. However, directly maintaining such accumulated covariance matrix would be computationally expensive, requiring $O(m^2 nT)$ time complexity. This is particularly challenging for transformer-based models where the parameter dimension $m, n$ are typically in the order of thousands.

To ensure computational efficiency, we use Frequent Directions (FD) [Ghashami et al., 2016, Wan and Zhang, 2018, 2022], a deterministic matrix sketching algorithm, to maintain a low-rank approximation of streaming gradients. The FD algorithm processes the gradients sequentially while providing a guarantee on approximation quality [Wan and Zhang, 2021, Yang et al., 2025]. Specifically, we first compute a low-rank matrix $Q_{\tau,t} \in \mathbb{R}^{m \times r_2}$ with $r_2 \ll n$ through singular value decomposition (SVD):

$$\begin{aligned} U\Sigma V^{\top} &= \mathrm{SVD}_{r_2}(G'_{\tau,t}) \\ Q_{\tau,t} &= U\Sigma. \end{aligned} \tag{6}$$

Here, $\mathrm{SVD}_{r_2}(\cdot)$ denotes a truncated SVD that retains the top-$r_2$ singular values. The resulting low-rank matrix $Q_{\tau,t}$ enables us to efficiently approximate the gradient covariance matrix:

$$Q_{\tau,t} Q_{\tau,t}^{\top} \approx G'_{\tau,t} G'^{\top}_{\tau,t}. \tag{7}$$

Based on this approximation, we further compute a sketch matrix $S_{\tau,t} \in \mathbb{R}^{m \times r_2}$ that incrementally consolidates the gradient covariance information from all training steps up to step $t$. This consolidation is achieved by combining the previous sketch matrix $S_{\tau,t-1}$ with the current approximation $Q_{\tau,t}$. The update of $S_{\tau,t}$ is as follows:

$$\begin{aligned} U'\Sigma'V'^{\top} &= \mathrm{SVD}_{r_2}([S_{\tau,t-1}, Q_{\tau,t}]) \\ S_{\tau,t} &= U'\sqrt{\Sigma'^2 - \sigma_t I_{r_2}}, \sigma = \Sigma'^2_{r_2,r_2}. \end{aligned} \tag{8}$$

We initialize $S_{\tau,1} = Q_{\tau,1}$, and after $T$ iterations, we obtain the final task-specific sketch matrix $S_{\tau,T}$, which satisfies:

$$S_{\tau,T} S_{\tau,T}^{\top} \approx \sum_{t=1}^{T} G'_{\tau,t} G'^{\top}_{\tau,t}. \tag{9}$$

By analyzing the dominant singular vectors of $S_{\tau,T}$, we can effectively estimate the principal subspace of the current task. Note that we update $S_{\tau,t}$ every $K$ steps to match the update frequency of projection matrix $P_{\tau,t}$, ensuring consistency in our approximation process. The effectiveness of CoSO relies on the accuracy of low-rank approximation, which is formalized through the following Proposition 1.

**Proposition 1.** *Given a sequence of projected gradients $\{G'_{\tau,t}\}_{t=1}^{T}$ and low-rank matrix $\{Q_{\tau,t}\}_{t=1}^{T}$, where $G'_{\tau,t} \in \mathbb{R}^{m \times n}$ and $Q_{\tau,t} \in \mathbb{R}^{m \times r_2}$. The final sketch matrix is $S_{\tau,T} \in \mathbb{R}^{m \times r_2}$. Let $A = \sum_{t=1}^{T} G'_{\tau,t} G'^{\top}_{\tau,t}$, $\tilde{A} = \sum_{t=1}^{T} Q_{\tau,t} Q_{\tau,t}^{\top}$. For any $k < r_2$ the approximation error is bounded by:*

$$\|A - S_{\tau,T} S_{\tau,T}^{\top}\|_2 \le \sum_{t=1}^{T} \sigma_t^2 + \frac{\|\tilde{A} - [\tilde{A}]_k\|_F^2}{r_2 - k}, \tag{10}$$

*where $\sigma_t$ is the $(r_2 + 1)$-th singular value of $G'_{\tau,t}$ and $[\tilde{A}]_k$ is the minimizer of $\|\tilde{A} - [\tilde{A}]_k\|_F$ overall rank $k$ matrices.*

**Remark.** *Because the gradients often exhibit low-rank structure [Cosson et al., 2023, Zhao et al., 2024], their singular values decay rapidly. Consequently, the error $\sum_{t=1}^{T} \sigma_t^2$ would be negligibly small when $r_2$ exceeds the intrinsic rank of the gradients. By maintaining low-rank sketch matrix $S_{\tau,T}$, we reduce the cost of computing $\sum_{t=1}^{T} G'_{\tau,t} G'^{\top}_{\tau,t}$ from $O(m^2 nT)$ to $O(mnr_2 T)$, where $r_2 \ll m$. Proposition 1 ensure that our low-rank approximation captures the most significant directions in the gradient space. The error bound provides practical guidance for choosing the rank $r_2$: larger values lead to better approximation at the expense of additional computation and memory. To better preserve the task information, we set $r_2$ to be slightly larger than $r_1$, where $r_1$ is the projection rank introduced in Section 3.2. The proof is provided in Appendix A.*

**Update Orthogonal Basis Matrix.** Once the final task-specific sketch matrix $S_{\tau,T}$ is computed, we use it to update the orthogonal basis matrix $\mathcal{M}_{\tau-1}$ to incorporate the optimization subspace of the task $\tau$. First, we extract the principal directions of the current task by performing SVD on its sketch matrix:

$$U_\tau \Sigma_\tau V_\tau^{\top} = \text{SVD}(S_{\tau,T}). \tag{11}$$

Then, we determine the number of directions to retain based on the sum of squared singular values. Following the principle of matrix approximation with SVD, we select $k$ as the biggest value that satisfies:

$$\frac{\sum_{i=1}^{k} \sigma_i^2}{\sum_{j=1}^{r_2} \sigma_j^2} \leq \epsilon_{th}, \tag{12}$$

where $\epsilon_{th} \in (0,1]$ is a threshold hyperparameter controlling the ratio to preserve, and $\sigma_i$ is the $i$-th singular values in descending order. This criterion ensures that the selected $k$ directions capture at least $\epsilon_{th}$ fraction of the total variance in the gradient space. Finally, we expand the orthogonal basis matrix $\mathcal{M}_{\tau-1}$ by incorporating these new directions:

$$\mathcal{M}_\tau = [\mathcal{M}_{\tau-1}, U_\tau[:,: k]]. \tag{13}$$

The above selection and update process ensure that we capture the most important learning directions for each task while maintaining orthogonality between different tasks' subspaces.

Due to space constraints, the complete CoSO algorithm is presented in Appendix B.

# 4 Experiments

We conduct comprehensive experiments with varying numbers of sequential tasks to evaluate CoSO's effectiveness across multiple datasets. We first outline our experimental settings, then present detailed results and analyses.

## 4.1 Experimental Settings

**Datasets and Evaluation Metrics.** Following previous works [Wang et al., 2022b, Liang and Li, 2024], we evaluate CoSO on three widely-used continual learning benchmarks: ImageNet-R [Hendrycks et al., 2021], CIFAR100 [Krizhevsky, 2009], and DomainNet [Peng et al., 2019]. ImageNet-R contains 200 classes from ImageNet with artistic style variations. Similar to existing works [Smith et al., 2023, Liang and Li, 2024, Wu et al., 2025], we create three different splits of ImageNet-R: 5 tasks with 40 classes per task, 10 tasks with 20 classes per task, and 20 tasks with 10 classes per task. For CIFAR100, we divide it into 10 tasks, each containing 10 classes. DomainNet consists of 345 classes across six distinct domains and is split into 5 tasks, with 69 classes per task.

We evaluate our method using two complementary metrics that are widely adopted in existing continual learning methods [Wang et al., 2022b, Liang and Li, 2024, Wu et al., 2025]. The first metric is the final accuracy $ACC_T$, which evaluates the model's overall performance across all tasks after the complete training process. The second metric is the average accuracy $\overline{ACC}_T$, which measures the model's learning stability throughout the training sequence and is calculated as $\overline{ACC}_T = \frac{1}{T} \sum_{i=1}^{T} ACC_i$, where $T$ denotes the total number of tasks. These two metrics capture both the model's ability to learn new tasks and retain knowledge of previously learned tasks, providing a comprehensive assessment of continual learning performance.

Table 1: Results (%) on ImageNet-R with varying numbers of tasks (5, 10 and 20). All reported results with mean and standard deviation are computed over 3 independent runs.

| Method | ImageNet-R (5 Tasks) | | ImageNet-R (10 Tasks) | | ImageNet-R (20 Tasks) | |
|---|---|---|---|---|---|---|
| | $ACC_5$ | $\overline{ACC}_5$ | $ACC_{10}$ | $\overline{ACC}_{10}$ | $ACC_{20}$ | $\overline{ACC}_{20}$ |
| L2P | $65.03_{\pm0.03}$ | $69.97_{\pm0.15}$ | $62.87_{\pm0.72}$ | $68.90_{\pm0.58}$ | $58.64_{\pm0.34}$ | $65.57_{\pm0.35}$ |
| DualPrompt | $68.24_{\pm0.23}$ | $71.82_{\pm0.39}$ | $65.30_{\pm0.52}$ | $69.62_{\pm0.29}$ | $60.47_{\pm0.54}$ | $65.91_{\pm0.52}$ |
| CODA-P | $73.65_{\pm0.15}$ | $77.88_{\pm0.30}$ | $72.10_{\pm0.29}$ | $76.90_{\pm0.41}$ | $67.16_{\pm0.11}$ | $72.34_{\pm0.44}$ |
| InfLoRA | $77.53_{\pm0.30}$ | $82.24_{\pm0.11}$ | $74.43_{\pm0.31}$ | $80.50_{\pm0.06}$ | $70.30_{\pm0.14}$ | $77.04_{\pm0.06}$ |
| SD-LoRA | $79.15_{\pm0.20}$ | $83.01_{\pm0.42}$ | $77.34_{\pm0.35}$ | $82.04_{\pm0.24}$ | $75.26_{\pm0.37}$ | $80.22_{\pm0.72}$ |
| VPT-NSP$^2$ | $79.72_{\pm0.19}$ | $84.33_{\pm0.29}$ | $77.87_{\pm0.10}$ | $83.09_{\pm0.26}$ | $75.42_{\pm0.27}$ | $81.32_{\pm0.21}$ |
| CoSO | $\mathbf{82.10}_{\pm0.13}$ | $\mathbf{86.38}_{\pm0.07}$ | $\mathbf{81.10}_{\pm0.39}$ | $\mathbf{85.56}_{\pm0.13}$ | $\mathbf{78.19}_{\pm0.28}$ | $\mathbf{83.69}_{\pm0.12}$ |

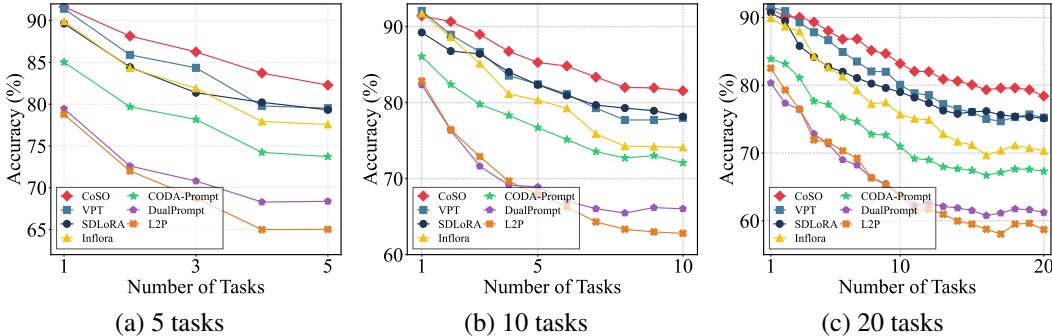

(a) 5 tasks     (b) 10 tasks     (c) 20 tasks

Figure 2: The detailed performance during the learning of ImageNet-R on (a) 5 tasks, (b) 10 tasks, and (c) 20 tasks.

**Baselines and Implementation Details.** We compare CoSO with several state-of-the-art PEFT-based methods: L2P [Wang et al., 2022b], DualPrompt [Wang et al., 2022a], CODA-Prompt (CODA-P) [Smith et al., 2023], InfLoRA [Liang and Li, 2024], VPT-NSP$^2$ [Lu et al., 2024], and SD-LoRA [Wu et al., 2025]. Comparing against both prompt-based and LoRA-based methods allows us to comprehensively evaluate the effectiveness of CoSO. In addition to the ViT-B/16 [Dosovitskiy et al., 2021] pretrained on ImageNet-1K, we also evaluate a self-supervised ViT-B/16 obtained with DINO [Caron et al., 2021]. Details of the experimental setup are provided in Appendix C.

## 4.2 Experimental Results

We evaluate CoSO against state-of-the-art continual learning methods across different experimental settings. Table 1 shows the performance comparison on ImageNet-R under various task partitions (5, 10, and 20 tasks). Across all partitions, CoSO delivers the highest final accuracy ($ACC$) and average accuracy ($\overline{ACC}$), confirming its robustness to mitigate forgetting. For the most challenging setting (20 tasks), CoSO attains 78.19% final accuracy and 83.69% average accuracy, while the best baseline method achieves 75.42% and 81.32%, respectively. For the ImageNet-R 10 tasks scenario, CoSO improves the final accuracy by 3.23% and the average accuracy by 2.47% compared to the best baseline method. Likewise, in the 5 tasks setting, CoSO still leads by 2.38% in final accuracy and 2.05% in average accuracy. This margin highlights CoSO's exceptional resistance to forgetting and its strong capacity to integrate new knowledge without eroding prior learning.

Figure 2 illustrates the evolution of accuracy throughout the continual learning process for various methods evaluated on ImageNet-R. It is evident that CoSO consistently maintains superior performance relative to other approaches, both during the intermediate phases and at the end of training. This ongoing superiority underscores CoSO's effectiveness in reducing interference from newly introduced tasks, resulting in a significantly slower decline in accuracy compared to competing methods. Complementary results in Table 2 reveal the same trend on CIFAR100 and DomainNet. On the DomainNet benchmark, CoSO outperforms the best baseline method by 1.75% in final accuracy and 1.37% in average accuracy, confirming its ability to generalize across heterogeneous visual domains.

Table 2: Results (%) on CIFAR100 (10 Tasks) and DomainNet (5 Tasks). All reported results with mean and standard deviation are computed over 3 independent runs.

| Method | CIFAR100 (10 Tasks) | | DomainNet (5 Tasks) | |
|---|---|---|---|---|
| | $ACC_{10}$ | $\overline{ACC}_{10}$ | $ACC_5$ | $\overline{ACC}_5$ |
| L2P | $82.64_{\pm 0.26}$ | $87.90_{\pm 0.19}$ | $70.03_{\pm 0.09}$ | $75.65_{\pm 0.06}$ |
| DualPrompt | $84.68_{\pm 0.22}$ | $90.12_{\pm 0.05}$ | $72.25_{\pm 0.05}$ | $77.84_{\pm 0.02}$ |
| CODA-P | $86.60_{\pm 0.37}$ | $91.46_{\pm 0.20}$ | $73.16_{\pm 0.07}$ | $78.75_{\pm 0.04}$ |
| InfLoRA | $86.85_{\pm 0.08}$ | $91.45_{\pm 0.16}$ | $73.09_{\pm 0.11}$ | $79.21_{\pm 0.08}$ |
| SD-LoRA | $87.30_{\pm 0.45}$ | $91.81_{\pm 0.27}$ | $73.20_{\pm 0.12}$ | $79.03_{\pm 0.04}$ |
| VPT-NSP$^2$ | $88.09_{\pm 0.12}$ | $92.48_{\pm 0.11}$ | $72.52_{\pm 0.13}$ | $78.68_{\pm 0.06}$ |
| CoSO | $\mathbf{88.77}_{\pm 0.16}$ | $\mathbf{92.99}_{\pm 0.23}$ | $\mathbf{74.27}_{\pm 0.07}$ | $\mathbf{80.05}_{\pm 0.04}$ |

Table 3: Ablation study results (%) on ImageNet-R with varying numbers of tasks (5, 10 and 20).

| Method | ImageNet-R (5 Tasks) | | ImageNet-R (10 Tasks) | | ImageNet-R (20 Tasks) | |
|---|---|---|---|---|---|---|
| | $ACC_5$ | $\overline{ACC}_5$ | $ACC_{10}$ | $\overline{ACC}_{10}$ | $ACC_{20}$ | $\overline{ACC}_{20}$ |
| w/o Orth | 79.35 | 85.22 | 75.90 | 83.43 | 69.75 | 78.88 |
| w/o FD | 80.72 | 85.44 | 78.83 | 84.45 | 76.68 | 82.41 |
| CoSO | 82.37 | 86.46 | 80.72 | 85.67 | 78.27 | 83.62 |

A detailed analysis of computational and memory costs are presented in Appendix D. The additional results with DINO [Caron et al., 2021] are provided in Appendix E.

**Ablation Study.** We conduct comprehensive ablation studies on ImageNet-R benchmark to validate the individual contributions of the orthogonal projection mechanism and the Frequent Directions (FD) based subspace consolidation. Specifically, we compare CoSO with two variants. The first variant (w/o Orth) removes the orthogonal projection, which directly uses the original gradients $G_{\tau,t}$ for optimization instead of the orthogonally projected gradients $G'_{\tau,t}$. This variant optimizes parameters in continuous subspaces without any orthogonality constraint, thereby ignoring task interference. The second variant (w/o FD) retains orthogonality but, instead of employing FD to consolidate all intermediate gradients from the current task, constructs the task-specific subspace using only the final subspace obtained at the end of that task.

The results are summarized in Table 3. Eliminating orthogonal projection (w/o Orth) leads to a sharp performance drop (8.52% in final accuracy) on 20 Tasks setting, highlighting the importance of excluding new gradients from the historical subspace to prevent interference. Replacing FD with the simplified strategy that builds each task-specific subspace from only the final gradient subspace (w/o FD) also degrades performance, lowering final accuracy by 1.65%, 1.89% and 1.59% for 5, 10 and 20 Tasks settings, respectively. This drop confirms that aggregating all intermediate gradients through incremental FD updates captures richer task information than using a single terminal subspace. Across the table, the full method delivers the highest final and average accuracies, indicating that both orthogonal projection and FD consolidation are indispensable for robust continual learning.

## 5 Conclusion

In this paper, we propose Continuous Subspace Optimization for Continual Learning (CoSO). CoSO optimizes the pre-trained models within continuous subspaces. By maintaining orthogonality between the current task's optimization subspace and that of historical tasks, CoSO effectively mitigates the interference. CoSO maintains a compact task-specific component while learning a task. After completing the current task, the task-specific component is used to update the historical task subspace. Extensive experiments on standard benchmarks demonstrate that CoSO consistently outperforms state-of-the-art baselines in both final accuracy and average accuracy over time, confirming its effectiveness and robustness across diverse data streams. In the future, a challenging open problem is to extend CoSO to multimodal task settings.

## Acknowledgments and Disclosure of Funding

This work was partially supported by National Science and Technology Major Project (2022ZD0114801), and NSFC (U23A20382).

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

## A  Proof of Proposition 1

Recall that $\{G'_{\tau,t}\}_{t=1}^{T} \subset \mathbb{R}^{m \times n}$ is the sequence of projected gradients for task $\tau$, $A_t = G'_{\tau,t} G'^{\top}_{\tau,t}$, $A = \sum_{t=1}^{T} G'_{\tau,t} G'^{\top}_{\tau,t}$, $\tilde{A}_t = Q_{\tau,t} Q^{\top}_{\tau,t}$, $\tilde{A} = \sum_{t=1}^{T} Q_{\tau,t} Q^{\top}_{\tau,t}$ and sketch matrix $S_{\tau,T} \in \mathbb{R}^{m \times r_2}$.

Because $Q_{\tau,t}$ is the rank $r_2$ approximation of $G'_{\tau,t}$, for every step $t$, we have

$$\|A_t - \tilde{A}_t\|_2 = \sigma_t^2, \tag{14}$$

where $\sigma_t$ is the $(r_2 + 1)$ singular value of $G'_{\tau,t}$.

Using the triangle inequality together with Eq. (14),

$$\|A - \tilde{A}\|_2 = \left\| \sum_{t=1}^{T} (A_t - \tilde{A}_t) \right\|_2 \leq \sum_{t=1}^{T} \|A_t - \tilde{A}_t\|_2 = \sum_{t=1}^{T} \sigma_t^2. \tag{15}$$

Since we use FD to compute $S_{\tau,T}$ based on $\{Q_{\tau,t}\}_{t=1}^{T}$, from Theorem 1.1 of Ghashami et al. [2016], we have

$$\|\tilde{A} - S_{\tau,T} S^{\top}_{\tau,T}\|_2 \leq \frac{\|\tilde{A} - [\tilde{A}]_k\|_F^2}{r_2 - k}, \tag{16}$$

where $[\tilde{A}]_k$ is the minimizer of $\|\tilde{A} - [\tilde{A}]_k\|_F$ overall rank $k$ matrices. Applying the triangle inequality to $\|A - S_{\tau,T} S^{\top}_{\tau,T}\|$ and substituting Eq. (15) and (16) gives

$$\|A - S_{\tau,T} S^{\top}_{\tau,T}\|_2 \leq \|A - \tilde{A}\|_2 + \|\tilde{A} - S_{\tau,T} S^{\top}_{\tau,T}\|_2$$
$$\leq \sum_{t=1}^{T} \sigma_t^2 + \frac{\|\tilde{A} - [\tilde{A}]_k\|_F^2}{r_2 - k}, \tag{17}$$

which is exactly (10).

## B  CoSO Algorithm

We present the the detailed procedure in Algorithm 1.

## C  Experimental Setups and Implementation Details

Following existing works [Smith et al., 2023, Wu et al., 2025], we adopt ViT-B/16 [Dosovitskiy et al., 2021] pre-trained on ImageNet-21K and fine-tuned on ImageNet-1K as our backbone model, which consists of 12 transformer blocks. For fair comparison, all methods use the same ViT-B/16 backbone and optimizer. Additionally, we also evaluate a self-supervised ViT-B/16 obtained with DINO [Caron et al., 2021]. The optimization is performed using Adam [Kingma, 2014] optimizer with $\beta_1 = 0.9$ and $\beta_2 = 0.999$. The training epochs vary across datasets: 40 epochs for ImageNet-R, 20 epochs for CIFAR100, and 5 epochs for DomainNet. We maintain a consistent batch size of 128 across all experiments. Results are averaged over 3 independent runs, and we report the corresponding standard deviation. Notably, CoSO only optimize the output projection layers in multi-head attention module rather than QKV transformations.

We present the detailed hyperparameter settings of CoSO in Table 4. These hyperparameters are carefully tuned to balance memory efficiency and performance, reflecting the varying complexity of the datasets. The hyperparameter settings of baseline methods are following existing work [Wang et al., 2022a, Smith et al., 2023, Liang and Li, 2024, Lu et al., 2024, Wu et al., 2025]. For all datasets, we employ minimal data augmentation, consisting of random resized cropping to $224 \times 224$ pixels and random horizontal flipping during training, without any additional augmentation techniques. To prevent overfitting, we followed VPT-NSP[2] [Lu et al., 2024], setting the temperature parameter in the cross-entropy loss to 3 for all datasets. All experiments were conducted on NVIDIA A6000 GPUs with 48GB memory using PyTorch 2.5.1.

The projection rank ($r_1$) determines the dimensionality of the low-rank subspace for gradient projection. For simpler datasets like CIFAR100, a lower value of $r_1 = 15$ is sufficient, while more

---
**Algorithm 1** CoSO for Continual Learning
---
1: **Input:** A layer weight matrix $W \in \mathbb{R}^{m \times n}$, step size $\eta$, decay rates $\beta_1, \beta_2$, projection rank $r_1$, FD rank $r_2$, threshold $\epsilon$ and update gap $K$.
2: Initialize first-order moment $M_0 \in \mathbb{R}^{m \times r} \leftarrow 0$
3: Initialize second-order moment $V_0 \in \mathbb{R}^{m \times r} \leftarrow 0$
4: Initialize sketch matrix $S_{\tau,0} \in \mathbb{R}^{m \times r} \leftarrow 0$
5: Initialize orthogonal projection matrix $\mathcal{M}_0 \leftarrow 0$
6: **for** Task $\tau \in 1 \ldots N$ **do**
7:     **for** step $t \in 1 \ldots T$ **do**
8:         $G_{\tau,t} \leftarrow \nabla_{W_{\tau,t}} L(W_{\tau,t})$                  ▷ Compute mini-batch gradient for task $\tau$
9:         $G'_{\tau,t} \leftarrow G_{\tau,t} - \mathcal{M}_{\tau-1} \mathcal{M}_{\tau-1}^\top G_{\tau,t}$             ▷ Orthogonal projection
10:         **if** $t \bmod K == 0$ **then**
11:             $U\Sigma V^\top = \mathrm{SVD}(G'_{\tau,t})$
12:             $P_{\tau,t} = U[:, : r_1]$                  ▷ Compute projection matrix $P_{\tau,t}$
13:             Update $S_{\tau,t}$ through Eq. (6) and (8)    ▷ Use FD to consolidate gradient information
14:         **else**
15:             $P_{\tau,t} \leftarrow P_{\tau,t-1}$
16:             $S_{\tau,t} \leftarrow S_{\tau,t-1}$
17:         **end if**
18:         $R_{\tau,t} \leftarrow P_{\tau,t}^\top G'_{\tau,t}$            ▷ Project orthogonal gradient into low rank space
19:         Use $R_{\tau,t}$ to compute $N_{\tau,t}$ through Eq. (4)          ▷ Update $R_{\tau,t}$ by Adam
20:         $\tilde{G}_{\tau,t} \leftarrow P_{\tau,t} N_{\tau,t}$              ▷ Project gradient back to original space
21:         $W_{\tau,t} \leftarrow W_{\tau,t-1} - \eta \cdot \tilde{G}_{\tau,t}$
22:     **end for**
23:     Update the historical subspaces basis matrix $\mathcal{M}_{\tau-1}$ through Eq. (11), (12) and (13)
24: **end for**
---

Table 4: Hyperparameter settings for different datasets.

| Hyperparameter | CIFAR100 | ImageNet-R | DomainNet |
|---|---|---|---|
| Projection rank ($r_1$) | 15 | 50 | 70 |
| Frequent directions rank ($r_2$) | 100 | 120 | 160 |
| Update gap ($K$) | 1 | 1 | 20 |
| Threshold ($\epsilon_{th}$) | 0.98 | 0.98 | 0.98 |

complex datasets such as ImageNet-R and DomainNet require higher values ($r_1 = 50$ and $r_1 = 70$, respectively) to capture a richer set of gradient directions. The Frequent Directions rank ($r_2$) is consistently set higher than $r_1$ across all datasets. This design choice ensures that CoSO can capture a broader range of directions, reducing information loss during continual learning. As the dataset complexity increases, $r_2$ is adjusted upward to retain more task information.

The update gap K is adjusted based on the characteristics of each dataset. For DomainNet, we use a larger update gap ($K = 20$) due to its larger and more diverse task structure, where frequent updates may become redundant. In contrast, CIFAR100 and ImageNet-R exhibit rapid gradient changes, necessitating a smaller $K$. Finally, the threshold ($\epsilon$) is uniformly set to 0.98 across all datasets. This value is selected to maintain a high retention rate of gradient information within the subspace.

## D   Analysis of Computational and Memory Costs

We conducted a comparative analysis of CoSO and baseline methods with respect to computational cost (reported as estimated GFLOPs) and memory usage, as summarized in Table 5. CoSO requires half the computational cost of prompt-based methods (such as L2P, DualPrompt, and CODA-P), as it avoids the need for twice forward passes through the network. In terms of memory usage, CoSO is on par with other low-rank adaptation techniques such as InfLoRA (13.44). Its slightly higher memory footprint (13.61) stems from using a larger rank for gradient subspace approximation, which enables better capture of task-specific patterns and leads to superior performance. Notably,

Table 5: Comparison on ImageNet-R (10 Tasks) in terms of computation (GFLOPs) and memory usage.

| Method | GFLOPs | Memory Usage (G) |
|---|---|---|
| L2P | 70.24 | 12.90 |
| DualPrompt | 70.24 | 12.96 |
| CODA-P | 70.24 | 12.97 |
| InfLoRA | 35.12 | 13.44 |
| SD-LoRA | 35.12 | 15.62 |
| VPT-NSP$^2$ | 35.83 | 11.54 |
| CoSO | 35.12 | 13.61 |

Table 6: Results (%) on ImageNet-R (10 Tasks). All reported results with mean and standard deviation are computed over 3 independent runs.

| Method (DINO) | ImageNet-R (10 Tasks) | |
|---|---|---|
| | $ACC_{10}$ | $\overline{ACC}_{10}$ |
| L2P | $61.94_{\pm 0.45}$ | $68.77_{\pm 0.27}$ |
| DualPrompt | $60.40_{\pm 0.18}$ | $67.65_{\pm 0.07}$ |
| CODA-P | $64.63_{\pm 0.33}$ | $72.20_{\pm 0.30}$ |
| InfLoRA | $67.91_{\pm 0.23}$ | $76.40_{\pm 0.03}$ |
| SD-LoRA | $69.78_{\pm 0.63}$ | $65.73_{\pm 0.35}$ |
| VPT-NSP$^2$ | $69.68_{\pm 0.20}$ | $77.24_{\pm 0.16}$ |
| CoSO | $\mathbf{71.60}_{\pm 0.44}$ | $\mathbf{79.28}_{\pm 0.16}$ |

simply increasing the rank for InfLoRA would not yield similar improvements, as its performance is limited by the constraint of fixed subspaces. Compared with SD-LoRA (15.62), which incurs the greatest memory overhead, CoSO offers a more efficient alternative while delivering competitive performance. Overall, these results highlight CoSO's ability to strike a favorable balance between computational efficiency and memory usage, making it a scalable solution for continual learning across diverse tasks.

# E    Additional Experiment Results on ImageNet-R

To further verify CoSO's generality, we test it on a self-supervised ViT-B/16 backbone trained with DINO [Caron et al., 2021] on ImageNet-R (10 Tasks). The results are presented in Table 6. CoSO outperforms the best baseline method with a considerable margin, confirming its ability to generalize across various vision transformers.

