# OpenReview forum: "Continuous Subspace Optimization for Continual Learning"
_NeurIPS.cc/2025/Conference — NeurIPS 2025 poster_

### Official Review · Reviewer_HjST · 2025-06-19

**Clarity:** 3
**Significance:** 3
**Originality:** 3
**Rating:** 4
**Confidence:** 4

**Summary:**

The authors propose CoSO, a continual learning method for adapting foundational vision models. It fine-tunes pretrained models within continuous subspaces derived from the SVD of gradients. To mitigate interference, the method enforces orthogonality between the optimization subspace of the current task and those of previous tasks. After completing a task, its task-specific component is accumulated to form a historical subspace, which is used in subsequent tasks. Experiments on several continual learning benchmarks demonstrate that CoSO outperforms recent baselines such as SD-LoRA.

**Questions:**

1. Please clarify why strict orthogonality (i.e., no interference) is necessary. Could shared representations benefit learning in some cases?
2. Please elaborate on how CoSO differs from SD-LoRA in design, efficiency, and empirical performance.
3. Please highlight the technical novelty of CoSO beyond combining known techniques.
4. Please include more detailed evaluation to show how well CoSO mitigates forgetting and addresses stability-plasticity.
5. SD-LoRA includes insightful empirical analyses, such as demonstrating the importance of previously learned directions for the current task. Could the authors provide similar analyses to show the role of historical subspaces in CoSO?

**Ethical Concerns:**

["NO or VERY MINOR ethics concerns only"]

**Final Justification:**

The authors clarify that the orthogonality constraint in CoSO is not meant to enforce complete independence from past knowledge, but rather to mitigate harmful gradient interference during optimization. They also clarify the distinctions between CoSO and SD-LoRA, both of which are for continual learning. The authors then provide additional experimental results on balancing stability and plasticity by showing Backward Transfer (BWT). Thus, I maintain my positive score.

**Quality:**

3

**Strengths And Weaknesses:**

**Strengths:**

1. Continual learning on foundational models is an emerging and important research area with practical significance, yet it remains underexplored.
2. The algorithm is simple yet effective and grounded in solid theoretical analyses (Propositions 1 and 2).
3. CoSO shows strong empirical performance, outperforming recent baselines such as SD-LoRA on vision tasks.

**Weaknesses:**

1. While enforcing orthogonality between task-specific subspaces reduces interference, it is unclear why complete separation is necessary. Is it always desirable for new task updates to be entirely independent of prior knowledge? Could past knowledge be beneficial for the current task? If strict independence is required, why not simply train a separate model per task?
2. CoSO and SD-LoRA appear to share several similarities. A more detailed comparison is needed, including algorithmic rationale and computational efficiency. For instance, CoSO (1) assigns a distinct subspace to each task and (2) leverages previously learned subspaces during training—both also present in SD-LoRA through task-specific LoRA adapters. Clarification on how CoSO differs and what drives its empirical improvements would be valuable.
3. The method builds on existing techniques such as SVD, GaLore, and FD, with orthogonality as the key addition for continual learning. A more thorough explanation of its technical novelty would strengthen the paper.
4. All evaluation metrics are averaged over current and past tasks. More fine-grained analysis would help assess the method’s ability to mitigate forgetting and maintain the stability-plasticity trade-off.

---

> ### Author Rebuttal · Authors · 2025-07-28
>
> We sincerely appreciate your constructive feedback. We have carefully considered your concerns and our responses are provided as follows.
>
> ---
> **W1 & Q1.** It is unclear why complete separation is necessary. Is it always desirable for new task updates to be entirely independent of prior knowledge? Could past knowledge be beneficial for the current task? If strict independence is required, why not simply train a separate model per task? Please clarify why strict orthogonality (i.e., no interference) is necessary. Could shared representations benefit learning in some cases?
>
> **A1.** We thank the reviewer for raising this important point regarding the necessity of orthogonal subspaces. We clarify that the orthogonality constraint in CoSO is not meant to enforce complete independence from past knowledge, but rather to mitigate harmful gradient interference during optimization. By projecting the current task's gradients orthogonally to the subspaces important for previous tasks, CoSO prevents overwriting past knowledge while still allowing parameter updates in complementary directions.
>
> Notably, the orthogonality constraint is applied only during optimization. CoSO maintains a single shared pre-trained model throughout the entire training process. During inference phase, all model parameters jointly contribute, ensuring that useful representations from earlier tasks are retained and leveraged in future predictions. Therefore, past knowledge can benefit the current task.
>
> We note that our setting follows the task-agnostic inference protocol, where the task label is unavailable at test time. In this setting, even if a separate model were trained per task, it would be infeasible to determine which model to use at inference. In contrast, CoSO maintains a single unified model that achieves strong performance without requiring any task labels at test time, demonstrating its practical effectiveness and scalability under realistic continual learning scenarios.
>
> ---
> **W2 & Q2.** CoSO and SD-LoRA appear to share several similarities. A more detailed comparison is needed, including algorithmic rationale and computational efficiency. Please elaborate on how CoSO differs from SD-LoRA in design, efficiency, and empirical performance.
>
> **A2.**  We appreciate the reviewer's insightful observation and agree that both CoSO and SD-LoRA share the common goal of continual learning through parameter-efficient fine-tuning. However, the two approaches differ in both design and implementation, as clarified below:
>
> **(1) Method Design:**
> SD-LoRA assigns a separate LoRA adapter to each task, where each adapter comprises a pair of low-rank matrices. While SD-LoRA decouples magnitude and direction to enable flexible adaptation, it does not explicitly prevent task interference. In contrast, CoSO avoids task-specific modules by optimizing a shared backbone. To mitigate interference, CoSO enforces gradient orthogonality, ensuring updates remain orthogonal to prior task subspaces.
>
> **(2) Memory Efficiency:**
> CoSO leverages gradient-based low-rank projection to optimize a single shared backbone, enabling memory-efficient training without maintaining multiple sets of adapter weights. This optimization strategy ensures that the GPU memory consumption remains constant, regardless of the number of tasks. In contrast, SD-LoRA must retain all previously learned adapters in memory during training to prevent forgetting, resulting in the GPU memory usage that grows linearly with the number of tasks. This becomes a bottleneck when handling long task sequences. As shown in Appendix E (Table 5), CoSO consumes less memory than SD-LoRA, demonstrating better scalability in both theory and practice.
>
> **(3) Empirical Performance:**
> CoSO consistently outperforms SD-LoRA across all benchmarks. For example, on ImageNet-R (20 tasks), CoSO improves final accuracy by 2.93% and average accuracy by 3.47% (Table 1). These results highlight the effectiveness of gradient space orthogonalization and dynamic subspace composition in enhancing knowledge retention and adaptation.
>
> In summary, CoSO differs from SD-LoRA by (i) using online gradient-derived subspaces rather than static adapters, (ii) explicitly preventing forgetting via orthogonal projection, and (iii) requiring less memory. We will revise the paper to make these distinctions clearer.
>
> ---
> **W3 & Q3.** The method builds on existing techniques such as SVD, GaLore, and FD, with orthogonality as the key addition for continual learning. A more thorough explanation of its technical novelty would strengthen the paper. Please highlight the technical novelty of CoSO beyond combining known techniques.
>
> **A3.** We thank the reviewer for raising this important point. We would like to clarify the high-level motivation behind CoSO and its key technical contributions.
>
> Recent continual learning methods with pre-trained models often follow a common paradigm: adapting a fixed subset of parameters (e.g., prompts or low-rank adapters) to mitigate forgetting while preserving generalization ability of foundation models. Among them, LoRA-based approaches (e.g., InfLoRA) constrain updates within a fixed low-rank subspace. While this reduces interference, it also limits the model's learning capacity.
>
> Our work takes a novel perspective: instead of introducing prompts or adapters, we tackle the limitation of learning capacity by rethinking the optimization space. Our motivation is to fully exploit the expressive potential of pre-trained models by approximating full-rank fine-tuning in a parameter-efficient way, without modifying the model architecture. To this end, we propose to optimize in a sequence of evolving subspaces following GaLore, each derived from current gradients via truncated SVD. This dynamic subspace formulation allows CoSO to retain adaptation flexibility while remaining memory efficient.
>
> However, unconstrained subspace evolution across tasks leads to severe forgetting. To address this, we introduce a gradient orthogonality constraint that ensures updates for each task do not interfere with learned knowledge. A major technical challenge is that each task comprises multiple subspaces across training steps. To make orthogonalization feasible, we aggregate these local subspaces into a compact component using Frequent Directions, which allows us to track only the essential directions unique to each task.
>
> We would like to emphasize that CoSO is not a simple combination of SVD, GaLore, and FD. Each component plays a distinct role in addressing specific challenges of continual learning. Together, they form a cohesive framework that improves both learning capacity and knowledge retention. As a result, CoSO consistently outperforms existing Prompt-based and LoRA-based methods across benchmarks.
>
> ---
> **W4 & Q4.** More fine-grained analysis would help assess the method's ability to mitigate forgetting and maintain the stability-plasticity trade-off. Please include more detailed evaluation to show how well CoSO mitigates forgetting and addresses stability-plasticity.
>
> **A4.** We appreciate the reviewer's suggestion to include more detailed evaluation. Our primary evaluation focuses on final task accuracy and average accuracy, which are widely adopted in existing PEFT-based continual learning works. These metrics provide a comprehensive view of both adaptation (learning new tasks) and retention (preserving previous knowledge), and allow for direct comparison with prior state-of-the-art methods.
>
> Following the suggestion of reviewer fJcc, we have computed Backward Transfer (BWT) for all methods on ImageNet-R 20 tasks to further evaluate the effectiveness of mitigating catastrophic forgetting. The results below show that CoSO not only achieves the highest final and average accuracy, but also obtains the highest BWT (closest to zero), indicating minimal forgetting.
>
> | Method     | IMNR-20     | BWT (IMNR-20) |
> | ---------- | ----------- | ------------- |
> | L2P        | 58.64/65.57 | -6.89         |
> | DualPrompt | 60.47/65.91 | -4.59         |
> | CODA-P     | 67.16/72.34 | -4.50         |
> | InfLoRA    | 70.30/77.04 | -7.46         |
> | VPT-NSP    | 75.42/81.32 | -7.32         |
> | SD-LoRA    | 75.26/80.22 | -6.39         |
> | CoSO       | 78.19/83.69 | -3.97         |
>
> ---
> **Q5.** SD-LoRA includes insightful empirical analyses, such as demonstrating the importance of previously learned directions for the current task. Could the authors provide similar analyses to show the role of historical subspaces in CoSO?
>
> **A5.** We thank the reviewer for this insightful point. SD-LoRA retains all task-specific adapters and is therefore able to directly analyze the importance of previously learned adapters by reusing them during training. In contrast, CoSO does not store the actual parameter updates or direction vectors, preserving only the historical subspace constructed via sketching. While this difference prevents us from conducting similar analysis as in SD-LoRA, we can design experiments to assess the contribution of the historical subspace.
>
> Specifically, our ablation study removes the orthogonality constraint, allowing updates to overlap with past subspaces. The results show a substantial performance drop (e.g., −8.52% final accuracy in the 20-task setting), indicating that the historical subspace plays a crucial role in preserving past knowledge. This result strongly suggests that the accumulated subspace effectively mitigates interference.

---

### Official Review · Reviewer_ndbX · 2025-07-01

**Clarity:** 3
**Significance:** 3
**Originality:** 3
**Rating:** 5
**Confidence:** 4

**Summary:**

This paper introduces CoSO, a continual learning method that leverages low-rank gradient subspaces to fine-tune pre-trained models. It dynamically adapts a model by optimizing within a series of low-rank subspaces. These subspaces are derived from gradients via SVD and are kept orthogonal to historical task subspace to reduce interference. CoSO incrementally updates a task-specific subspace using the Frequent Directions. Experiments on benchmark datasets show that CoSO outperforms state-of-the-art methods.

**Questions:**

- Proposition 1 states that $G^\prime$ lies in the null space of $X$. However, in practice, $W$ is updated using $\tilde{G}$ in Eq.(7). Does Eq.(7) still satisfy Proposition 1? To avoid confusion, the authors should provide a clear explanation.

- Why does CoSO restrict updates to the linear projection layers and not to FFN parameters? Could this approach be extended to update the MLP layers in FFN as well?

- Could CoSO be applied to non-vision tasks like NLP? If so, what modifications would be needed for it to work effectively?

- Could CoSO be combined with other continual learning approaches, such as regularization-based or prototype-based?

**Ethical Concerns:**

["NO or VERY MINOR ethics concerns only"]

**Final Justification:**

The authors have provided a detailed and satisfactory response to my previous comments. Since my primary concerns have been adequately addressed, I confirm my positive rating of this manuscript.

**Limitations:**

Please refer to "Strengths And Weaknesses" and "Questions" above.

**Paper Formatting Concerns:**

Non

**Quality:**

3

**Strengths And Weaknesses:**

### Strengths
- The proposed CoSO extends low-rank adaptation by using dynamically evolving subspaces, allowing updates across multiple orthogonal subspaces. This design increases the model’s learning capacity, and the orthogonality explicitly reduces interference with previously learned knowledge.

- The authors incorporate Frequent Directions to incrementally capture and compress gradient information into a low-rank task-specific subspace. This strategy effectively reduces memory and computational costs while preserving the most significant update directions for each task.

- Comprehensive experiments across multiple datasets demonstrate that CoSO consistently outperforms baseline methods. The well-executed ablation studies effectively isolate the contributions of its key components (orthogonalization and FD).


### Weaknesses
- CoSO seems to involve quite a few hyperparameters. Specifically, $r_1$ and $r_2$ in particular appear very similar. Could the authors explain the specific difference between them, and whether they could potentially be merged into a single parameter?

- The baseline methods are mainly prompt-based and LoRA-based. It would be helpful if the authors include comparisons with a broader range of recent studies, such as RanPAC [1*], EASE [2*], etc.

Reference:

[1*] RanPAC: Random Projections and Pre-trained Models for Continual Learning. NeurIPS 2023.

[2*] Expandable Subspace Ensemble for Pre-Trained Model-Based Class-Incremental Learning. CVPR 2024.

---

> ### Author Rebuttal · Authors · 2025-07-28
>
> We sincerely appreciate your thoughtful and encouraging feedback on our work. Below, we provide a detailed response to your concern.
>
> ---
> **W1.** Explain the specific difference between $r_1$ and $r_2$, and whether they could potentially be merged into a single parameter?
>
> **R1.**  Thank you for the insightful question regarding $r_1$ and $r_2$. We fully agree that a model should ideally involve as few hyperparameters as possible to promote simplicity and ease of tuning. While these two parameters may appear similar, they are designed to serve different purposes.
>
> The projection rank $r_1$ is used during the optimization process to control the dimensionality of the gradient subspace in which parameter updates are performed. A smaller $r_1$ ensures that the projected gradient is memory-efficient and computationally light, which is especially critical during training.
>
> On the other hand, the Frequent Directions rank $r_2$ is used to construct the task-specific subspace by consolidating gradient directions across training steps. This subspace serves as a compact summary of the current task's critical update directions and is appended to the historical task subspace to preserve prior knowledge. To achieve a more accurate estimation of the task-specific gradient subspace, $r_2$ is set larger than $r_1$, which allows it to capture a broader spectrum of gradient variation over time.
>
> Yes, we can merge $r_1$ and $r_2$ into a single parameter. But we find that tuning them separately leads to better performance in practice. A relatively larger $r_2$ becomes important when $r_1$ is small, as it enables better approximation of the task-specific subspace.
>
> ---
> **W2.** The baselines are limited to Prompt-based and LoRA-based methods. It would be helpful to include comparisons with recent studies, such as RanPAC and EASE.
>
> **R2.** Thank you for the valuable suggestion. Following your suggestion, we conducted additional experiments on ImageNet-R using the official implementations and matched settings. The results further demonstrate the superior performance of CoSO across diverse continual learning paradigms.
>
> | Methods | IMNR-5 | IMNR-10  | IMNR-20 |
> | ------- | ----------- | ------------ | ------------ |
> | EASE    | 77.53/82.51 | 76.95/82.29  | 75.75/81.96  |
> | RanPAC  | 79.54/83.85 | 78.04/83.35  | 75.03/80.98  |
> | CoSO    | 82.10/86.38 | 81.10/85.56  | 78.19/83.69  |
>
> Our study focuses primarily on continual learning with pre-trained vision transformers under the parameter-efficient fine-tuning (PEFT) paradigm. The baseline methods we originally included are state-of-the-art PEFT-based approaches widely adopted in recent literature, ensuring a fair and relevant comparison in this specific setting. While RanPAC and EASE are valuable prototype-based methods that operate in the feature space via class prototypes, they follow a different paradigm from PEFT-based approaches.
>
> ---
> **Q1.** Does Eq.(7) still satisfy Proposition 1?
>
> **A1.** Thank you for the thoughtful question. Yes, Equation (7) still satisfies Proposition 1. In CoSO, we first project the raw gradient $G_{\tau,t}$ into the null space of the historical task subspace, yielding $G_{\tau,t}^{\prime}$. All subsequent steps in Equation (7), including low-rank projection, Adam update, and projection back, operate within the subspace spanned by $G_{\tau,t}^{\prime}$. As a result, the final update $\tilde{G}_{\tau,t}$ still satisfies Proposition 1.
>
> ---
> **Q2.** Why does CoSO restrict updates to the linear projection layers and not to FFN parameters? Could this approach be extended to update the MLP layers in FFN as well?
>
> **A2.** Thank you for this question. Our goal is to achieve parameter-efficient fine-tuning by updating as few parameters as possible while maintaining strong performance. Following this principle, CoSO restricts optimization to the output projection layers and keeps all other parameters (including the FFN layers) frozen. By focusing on the projection layers, we achieve effective continual adaptation while significantly reducing computational and memory overhead. Theoretically, CoSO can be extended to other modules such as FFN layers.
>
> ---
> **Q3.** Could CoSO be applied to non-vision tasks like NLP? If so, what modifications would be needed for it to work effectively?
>
> **A3.** Thank you for this interesting question. While our current work focuses on vision tasks using pre-trained vision transformers, the idea underlying CoSO, which involves projecting gradients into continuous subspaces and enforcing orthogonality across tasks, is model-agnostic and applicable to standard transformer architectures. Consequently, the algorithm can be directly adapted to popular NLP models such as LLaMA by targeting the corresponding projection layers, where gradient-based subspace adaptation remains effective. This presents a promising direction for cross-modal continual learning. We intend to explore this extension in our future research.
>
> ---
> **Q4.** Could CoSO be combined with other continual learning approaches, such as regularization-based or prototype-based?
>
> **A4.** Thank you for the insightful question. CoSO is designed as a subspace-based optimization framework that operates at the level of parameter gradients and explicitly enforces orthogonality between tasks. Due to this design, it naturally overlaps with the objective of many regularization-based methods (e.g., EWC), which also aim to prevent interference through constrained updates. As a result, combining CoSO with regularization-based approaches may offer limited additional benefit.
>
> In contrast, prototype-based methods operate in the feature space and are conceptually orthogonal to CoSO. These methods, which store and use class prototypes to facilitate classification and knowledge retention, can be naturally integrated with CoSO by replacing or augmenting the classifier head while retaining CoSO for backbone optimization. This hybridization can potentially improve performance by leveraging both parameter-space and feature-space strategies.

---

> > ### Comment · Reviewer_ndbX · 2025-08-06
> >
> > The authors have provided a detailed and satisfactory response to my previous comments. Since my primary concerns have been adequately addressed, I confirm my positive rating of this manuscript.

---

> > > ### Author Response · Authors · 2025-08-06
> > >
> > > Thank you for your positive feedback. We sincerely appreciate your time and thoughtful review, and we are glad that our responses have addressed your concerns.

---

### Official Review · Reviewer_JZv6 · 2025-07-01

**Clarity:** 3
**Significance:** 3
**Originality:** 3
**Rating:** 4
**Confidence:** 4

**Summary:**

This method (CoSO) is established on pre-trained models to tackle class-incremental learning problem. To mitigate catastrophic forgetting, Coso updates model parameters in the direction orthogonal to the historical task subspace. To address the limitation of restricted parameter update in a fixed low-rank subspace, Coso fine-tunes the model in a series of low-rank subspaces rather than a single one.

**Questions:**

See Weaknesses.

**Ethical Concerns:**

["NO or VERY MINOR ethics concerns only"]

**Final Justification:**

The core idea of orthogonal gradient projection in this paper is similar to the prior literature PGP, but the specific implementation is distinct. In PGP, the update is performed on prompts since PGP is a prompt-based method. In contrast, this work updates parameters of the linear projection layers in multi-head attention module. Both methods address the class-incremental learning with the help of pre-trained models. I think this paper deserves my final score.

**Limitations:**

Yes.

**Paper Formatting Concerns:**

N/A.

**Quality:**

3

**Strengths And Weaknesses:**

Strengths:
This work leverages gradient projection to restrict parameters update for class-incremental learning with parameter-efficient finetuning. Experiments validate that CoSO consistently surpasses SOTA methods.

Weaknesses:
1. Prior works [1][2] leverage gradient orthogonal projection to aid visual prompt tuning, instead of low-rank adaptation, for class-incremental learning. Why does the CoSO outperform them in various continual benchmarks? Is the better performance of CoSO attributed to the utilization of LoRA?
2. Prior NSP[1] and PGP[2] both are plug-and-play methods, and can be applied to VPT-Seq, L2P, DualPrompt, CLIP, etc. The paper does not discuss whether the CoSO is plug-and-play.
3. In my opinion, the historical task subspace $M_{t-1}$ should span the representation subspace of all previous tasks, rather than the gradient subspace, in order to make the proposition 1 valid. But in this paper, authors consider $M_{t-1}$ as basis matrix of the gradient subspace.

[1] Visual prompt tuning in null space for continual learning. In NeurIPS, 2024

[2] Prompt Gradient Projection for Continual Learning. In ICLR, 2024.

---

> ### Author Rebuttal · Authors · 2025-07-28
>
> We sincerely thank you for your constructive feedback on our paper. We hope our responses adequately address the concerns you raised.
>
> ---
> **W1.** Why does the CoSO outperform NSP[1] and PGP[2] in various continual benchmarks? Is the better performance of CoSO attributed to the utilization of LoRA?
>
> **R1.** Thank you for raising this insightful question. While prior works such as NSP[1] and PGP[2] leverage orthogonal gradient projection to enhance prompt-based continual learning, CoSO differs in both its update scope and optimization strategy, leading to consistently stronger performance.
> - Prior methods[1, 2] constrain updates to the additional prompt tokens appended to the transformer, limiting the model's ability for task-specific adaptation. In contrast, CoSO directly updates the projection layers of the pre-trained vision transformer, which significantly expands the learning capacity of the model for continual adaptation.
> - In our baselines, we also include state-of-the-art LoRA-based continual learning methods such as InfLoRA [3] and SD-LoRA [4]. InfLoRA restricts parameter updates to a fixed subspace, which inherently limits flexibility. In contrast, CoSO performs optimization in a series of dynamically evolving gradient subspaces. These subspaces are explicitly enforced to be orthogonal to the historical task subspace, enabling CoSO to continually discover new learning directions while avoiding interference.
>
> To summarize, CoSO's superior performance stems from the novel design that jointly enables dynamic subspace exploration and orthogonal knowledge preservation, which are absent in prior Prompt-based or LoRA-based approaches.
>
> ---
> **W2.** The paper does not discuss whether the CoSO is plug-and-play.
>
> **R2.** Thank you for this valuable suggestion. CoSO is indeed a plug-and-play optimization strategy in continual learning pipelines. Unlike approaches that require model architectural augmentation (e.g., prompts or adapter), CoSO operates solely on the optimization procedure. It only requires access to the gradients of the parameters being optimized and performs low-rank projections and orthogonalization during the update. Therefore, CoSO can be directly integrated with other methods (e.g., prototype-based) that update a subset of backbone parameters without changing the model architecture. We thank the reviewer for encouraging us to make this clearer in the manuscript.
>
> ---
> **W3.** The historical task subspace $\mathcal{M}_{\tau-1}$ should span the representation subspace of all previous tasks, rather than the gradient subspace.
>
> **R3.** Thank you for this insightful question. We assume that both the feature and gradient subspaces lie in the same space and capture similar task-specific update directions. Working in the gradient subspace allows for a seamless integration with our low-rank projection-based optimization strategy. Specifically, we can directly apply orthogonality constraints to parameter updates without requiring access to intermediate feature representations. This effectively prevents interference by ensuring that current gradient updates are orthogonal to those from previous tasks. As a result, the risk of increasing the loss on prior tasks is minimized, thereby preserving previously learned knowledge.
>
> We will revise Section 3.2 to make the above rationale clearer. We thank the reviewer again for raising this point.
>
> ---
> **Reference**
>
> [1] Visual prompt tuning in null space for continual learning. NeurIPS, 2024
>
> [2] Prompt Gradient Projection for Continual Learning. ICLR, 2024.
>
> [3] InfLoRA: Interference-Free Low-Rank Adaptation for Continual Learning. CVPR 2024.
>
> [4] SD-LoRA: Scalable Decoupled Low-Rank Adaptation for Class Incremental Learning. ICLR 2025

---

> > ### Comment · Reviewer_JZv6 · 2025-08-05
> > **Response to Author Rebuttal**
> >
> > Thank the authors for providing the rebuttal. The rebuttal partially resolves my concerns. The following is more advice on the manuscript.
> > **W1**: I think authors should add the reference of PGP [2] in the related works, and compare it with the proposed method.
> > **W2**: Apart from the analysis, it’s better to provide more experiments results of applying the proposed method to other pre-trained models or PEFT methods.
> > **W3**: The assumption that both the feature and gradient subspaces lie in the same space and capture similar task-specific update directions should be explicitly provided in the revised manuscript.

---

> > > ### Author Response · Authors · 2025-08-05
> > >
> > > Thank you for your thoughtful response. We first provide brief clarifications below, and will carefully revise our paper as you suggested.
> > >
> > > ---
> > > **R1.** Thank you for the suggestion. We will include PGP [1] in the Related Work and incorporate additional experimental comparisons in the main results. We notice that PGP enforces orthogonality on prompt gradients, whereas CoSO performs dynamic low-rank subspace optimization directly on the backbone, enabling more flexible and effective continual adaptation.
> > >
> > > ---
> > > **R2.** Thank you for the suggestion. In addition to our current analysis, we will include new experimental results comparing CoSO with other PEFT methods such as PGP and EASE [2] to further validate its effectiveness. We also note that comparisons with the DINO backbone are provided in the appendix.
> > >
> > > ---
> > > **R3.** Thank you for the suggestion. We will carefully revise the manuscript to explicitly state the assumption that both the feature and gradient subspaces lie in the same space and capture similar task-specific update directions.
> > >
> > > ---
> > > **Reference**
> > >
> > > [1] Prompt Gradient Projection for Continual Learning. ICLR, 2024.
> > >
> > > [2] Expandable Subspace Ensemble for Pre-Trained Model-Based Class-Incremental Learning. CVPR 2024.

---

### Official Review · Reviewer_fJcc · 2025-07-06

**Clarity:** 3
**Significance:** 2
**Originality:** 3
**Rating:** 4
**Confidence:** 3

**Summary:**

This paper proposes a method called CoSO to address the catastrophic forgetting issue of pre-trained models in continual learning. Unlike existing approaches that typically constrain parameter updates within a fixed low-rank subspace, CoSO optimizes the model across a series of continuous, dynamically generated low-rank subspaces derived from gradient singular value decomposition (SVD), thereby enhancing the model's learning capability.

**Questions:**

1. Appendix D states that due to the non-linearity of Softmax, CoSO is not applied to the QKV transformation layers within the attention mechanism. Could you clarify what proportion of the total fine-tuned parameters are excluded as a result? Furthermore, how would you assess the performance loss incurred by not being able to enforce orthogonality on these critical layers?
2. Regarding the baseline PEFT methods in the experiments, was data replay employed during fine-tuning? Since data replay is a common technique for mitigating catastrophic forgetting, have the authors considered including a fine-tuning baseline with data replay for a more comprehensive comparison?
3. Is the current comparison against baseline methods fair? When compared to methods like InfLoRA, how much of CoSO's performance gain can be attributed to its continuous subspace optimization strategy versus its larger number of trainable parameters (given that r2 > r1)? Would you consider conducting additional experiments to compare performance under a similar total parameter budget, for instance, by increasing the rank of InfLoRA to match the effective rank of CoSO?

**Ethical Concerns:**

["NO or VERY MINOR ethics concerns only"]

**Final Justification:**

My initial review raised several concerns about the completeness and fairness of the experimental evaluation. The authors have addressed most of these concerns by providing new experimental results.

Specifically, the addition of the BWT metric, comparisons against replay-based methods, and fair-budget comparisons have strengthened the paper's empirical claims. The remaining limitation—the focus on vision tasks—is minor in comparison and has been properly acknowledged by the authors.

Given that the key experimental concerns have been addressed, the paper's technical contributions are clearer. I will therefore maintain my positive score.

**Limitations:**

yes

**Paper Formatting Concerns:**

No formatting issues.

**Quality:**

3

**Strengths And Weaknesses:**

## Strengths:
1. The paper is exceptionally well-written, logically structured, and easy to comprehend.
2. Inspired by GaLore, CoSO employs continual learning with continuous subspace optimization to address two key pain points of existing methods. It avoids the model capacity limitations of fixed subspaces and actively prevents forgetting through orthogonal projection, presenting a novel approach.


## Weaknesses:
1. CoSO introduces several key hyperparameters, including projection rank `r1`, FD rank `r2`, update interval `K`, and threshold `ε_th`. As shown in Table 4, these hyperparameters vary significantly across different datasets. Does this suggest that the method requires problem-specific tuning to achieve optimal performance?
2. Tables 1 and 2 report the method's performance in terms of final task ACC and AVG ACC. These metrics are indirect for assessing knowledge retention. Continual learning typically employs more direct measures like Backward Transfer (BWT). The absence of such metrics makes it difficult to precisely evaluate the effectiveness of mitigating catastrophic forgetting.
3. To maintain theoretical orthogonality, CoSO restricts its optimization scope to the output projection layer, excluding the QKV layers. Compared to PEFT methods that can update QKV layers, this may limit the model's adaptability and sacrifice potential performance improvements.
4. The experiments exclusively use datasets from the visual domain, such as ImageNet, the CIFAR series, and COCO for object detection. The study does not include tasks from other modalities, like natural language processing. This limits the method's generalizability within a universal fine-tuning paradigm.

---

> ### Author Rebuttal · Authors · 2025-07-28
>
> We sincerely appreciate your valuable feedback. We have carefully considered your concerns and provided a detailed explanation below.
>
> ---
> **W1.** CoSO introduces several key hyperparameters, including projection rank $r_1$, FD rank $r_2$, update interval $K$, and threshold $\epsilon_{th}$. These hyperparameters vary significantly across different datasets. Does this suggest that the method requires problem-specific tuning to achieve optimal performance?
>
> **R1.** We appreciate the reviewer's concern about hyperparameter sensitivity. In our experiments,  we fixed $\epsilon_{\text{th}}$ = 0.98 across all datasets. This setting is found to be robust in practice. The update interval K determines how frequently the projection and sketch matrices are computed. On larger datasets like DomainNet, we use larger $K=20$ to reduce overhead, while for smaller datasets with more volatile gradients, we use $K=1$ for better performance. Many existing PEFT-based continual learning methods also involve dataset-specific hyperparameters such as prompt length or LoRA rank. We followed the standard practice by tuning all hyperparameters on validation splits, ensuring fairness in comparison.
>
> To assess the impact of $r_1$ and $r_2$, we conducted sensitivity experiments varying their values on ImageNet-R. The results (final acc and average acc) below indicate that CoSO remains robust across a wide range of rank settings, with only minor performance fluctuations.
>
> | Method          | IMNR-5      | IMNR-10     | IMNR-20     |
> | --------------- | ----------- | ----------- | ----------- |
> | CoSO ($r_1=30$) | 82.17/86.06 | 80.52/85.40 | 77.90/83.03 |
> | CoSO ($r_1=40$) | 82.15/86.13 | 80.23/85.35 | 78.03/83.25 |
> | CoSO ($r_1=50$) | 82.10/86.38 | 81.10/85.56 | 78.19/83.69 |
> | CoSO ($r_1=60$) | 81.93/86.25 | 80.60/85.43 | 78.18/83.75 |
>
> | Method           | IMNR-5      | IMNR-10     | IMNR-20     |
> | ---------------- | ----------- | ----------- | ----------- |
> | CoSO ($r_2=80$)  | 81.63/86.01 | 80.50/85.44 | 78.22/83.62 |
> | CoSO ($r_2=100$) | 81.95/86.10 | 80.50/85.58 | 78.35/83.54 |
> | CoSO ($r_2=120$) | 82.10/86.38 | 81.10/85.56 | 78.19/83.69 |
> | CoSO ($r_2=140$) | 82.38/86.14 | 81.93/85.75 | 77.60/83.15 |
>
> ---
> **W2.** Continual learning typically employs more direct measures like Backward Transfer (BWT). The absence of such metrics makes it difficult to precisely evaluate the effectiveness of mitigating catastrophic forgetting.
>
> **R2.** We appreciate the reviewer's suggestion to include more direct metrics such as BWT. Our primary evaluation focuses on final accuracy and average accuracy, which are widely adopted in existing PEFT-based continual learning works. These metrics jointly reflect adaptation and retention, enabling fair comparisons with state-of-the-art methods.
>
> Following the reviewer's suggestion, we compute BWT for all methods on ImageNet-R 20 tasks to evaluate the effectiveness of mitigating forgetting. The results below show that CoSO achieves the highest BWT, indicating minimal forgetting.
>
> | Method     | BWT (IMNR-20) |
> | ---------- | ------------- |
> | L2P        | -6.89         |
> | DualPrompt | -4.59         |
> | CODA-P     | -4.50         |
> | InfLoRA    | -7.46         |
> | VPT-NSP    | -7.32         |
> | SD-LoRA    | -6.39         |
> | CoSO       | -3.97         |
>
> ---
> **W3.** To maintain theoretical orthogonality, CoSO restricts its optimization scope to the output projection layer, excluding the QKV layers. Compared to PEFT methods that can update QKV layers, this may limit the model's adaptability and sacrifice potential performance improvements.
>
> **R3.** We thank the reviewer for raising this point. Our goal is to achieve parameter-efficient fine-tuning by updating as few parameters as possible while maintaining strong performance. Following this principle, CoSO restricts optimization to the output projection layers and keeps all other parameters (including the QKV layers) frozen. To assess the effectiveness of this selective update strategy, we compare CoSO against baselines such as InfLoRA and SD-LoRA, which fine-tune the QKV layers. CoSO consistently outperforms these methods across multiple benchmarks. This suggests that tuning the projection layers alone is sufficient for effective adaptation, validating our design choice.
>
> ---
> **W4.** The experiments exclusively use datasets from the visual domain, such as ImageNet, the CIFAR series, and COCO for object detection. The study does not include tasks from other modalities, like natural language processing. This limits the method's generalizability within a universal fine-tuning paradigm.
>
> **R4.** We thank the reviewer for pointing out the scope of our experimental setup. While our current experiments focus on the visual domain, this choice is intentional to ensure a fair comparison with strong and well-established baselines in vision-based continual learning. Notably, the core mechanisms of CoSO, namely gradient-based subspace projection and orthogonality constraints, are not limited to the visual modality. These mechanisms are inherently general and can be extended to transformers used in NLP. As part of our ongoing work, we are actively exploring extensions to vision-language settings, including the adaptation of CoSO to CLIP-based models. However, such extensions involve non-trivial re-implementations and are beyond the scope of this initial study. We intend to explore this extension in our future research.
>
> ---
>
> **Q1.** Appendix D states that due to the non-linearity of Softmax, CoSO is not applied to the QKV transformation layers within the attention mechanism. Could you clarify what proportion of the total fine-tuned parameters are excluded as a result? Furthermore, how would you assess the performance loss incurred by not being able to enforce orthogonality on these critical layers?
>
> **A1.** We appreciate the reviewer's question regarding the exclusion of QKV layers. In ViT-B/16, the QKV layers account for approximately 25% of the total parameters. Theoretically, CoSO can be applied to the remaining 75% of parameters, including projection layers and FFN layers, which still comprise the majority of the transformer's components.
>
> As discussed in **R3**, updating more parameters does not always yield better results. By freezing most of the model parameters (including the QKV layers), we are able to better preserve and leverage the generalization capabilities of the pre-trained backbone, allowing CoSO to achieve strong performance with relatively few trainable parameters.
>
> ---
> **Q2.** Regarding the baseline PEFT methods in the experiments, was data replay employed during fine-tuning? Have the authors considered including a fine-tuning baseline with data replay for a more comprehensive comparison?
>
> **A2.** We thank the reviewer for raising the question regarding data replay. As discussed in Section 3.1, we focus on a rehearsal-free setting in which the model is not allowed to access or store data from previous tasks. This setting is widely considered more challenging, as it forgoes the benefits of revisiting past data. Our setup aligns with recent state-of-the-art PEFT-based continual learning methods. To ensure fair comparisons, all baselines included in our experiments also follow this rehearsal-free protocol.
>
> To address your concern, we conduct additional experiments on ImageNet-R of MEMO[1], a typical replay-based method. For a fair comparison, we use 40 images per class for replay. In addition, following reviewer ndbX's suggestion, we include RanPAC[2] and EASE[3], two representative methods that adopt prototype-based replay. These additional results further validate the superior performance of CoSO.
>
> | Methods | IMNR-5       | IMNR-10     | IMNR-20     |
> | ------- | ------------ | ----------- | ----------- |
> | MEMO    | 72.18/80.02  | 70.12/79.04 | 67.60/77.98 |
> | EASE    | 77.53/82.51  | 76.95/82.29 | 75.75/81.96 |
> | RanPAC  | 79.54/83.85  | 78.04/83.35 | 75.03/80.98 |
> | CoSO    | 82.10/86.38  | 81.10/85.56 | 78.19/83.69 |
>
> ---
> **Q3.** Is the current comparison against baseline methods fair? When compared to methods like InfLoRA, how much of CoSO's performance gain can be attributed to its continuous subspace optimization strategy versus its larger number of trainable parameters (given that r2 > r1)? Would you consider conducting additional experiments to compare performance under a similar total parameter budget, for instance, by increasing the rank of InfLoRA to match the effective rank of CoSO?
>
> **A3.** We appreciate the reviewer's insightful concern regarding fairness in baseline comparisons. We would like to clarify that the projection rank $r_1$ defines the rank of the gradient subspace, and does not add any extra trainable parameters to the model. The FD rank $r_2$ is not used to directly update model weights, but helps construct the task-specific subspace via the FD. The improved performance is stems from CoSO's ability to (1) optimize in multiple evolving subspaces rather than a fixed one, and (2) enforce orthogonality to mitigate task interference.
>
> To further address your concern, we conduct additional experiments where we increase the rank in InfLoRA to match the rank of CoSO. As discussed in Appendix E,  increasing the rank of InfLoRA would not yield improvements.
>
> | Method            | IMNR-5      | IMNR-10     | IMNR-20     |
> | ----------------- | ----------- | ----------- | ----------- |
> | InfLoRA (rank 50) | 75.45/81.63 | 70.95/79.40 | 64.57/73.89 |
> | InfLoRA (rank 30) | 76.52/82.43 | 72.30/79.87 | 68.98/76.06 |
> | InfLoRA (rank 10) | 77.53/82.24 | 74.43/80.50 | 70.30/77.04 |
> | CoSO              | 82.10/86.38 | 81.10/85.56 | 78.19/83.69 |
>
> ---
> **Reference**
>
> [1] A Model or 603 Exemplars: Towards Memory-Efficient Class-Incremental Learning. ICLR 2023.
>
> [2] RanPAC: Random Projections and Pre-trained Models for Continual Learning. NeurIPS 2023.
>
> [3] Expandable Subspace Ensemble for Pre-Trained Model-Based Class-Incremental Learning.
> CVPR 2024.

---

> > ### Comment · Reviewer_fJcc · 2025-08-06
> >
> > I appreciate the author's hard work in the rebuttal! After carefully reviewing the responses and supplementary materials you provided, I feel you've addressed most of my questions (e.g., W2, Q2, and Q3) very well. The addition of new experimental data and detailed explanations makes me feel the research is quite solid.
> >
> > However, I still feel the paper could improve on a few minor details:
> >
> > Experimental Scalability: The current experimental design is quite good (based on the CV field), but a detailed discussion of future applications of the method to other fields (e.g., NLP) would have enriched the paper.
> >
> > I will keep the positive score.

---

> > > ### Author Response · Authors · 2025-08-06
> > >
> > > We sincerely appreciate your constructive feedback and your decision to maintain the positive score. We're glad that our responses have addressed most of your concerns.
> > >
> > > Following your valuable suggestion, we will include a detailed discussion on extending our method to other domains (e.g., NLP) in the revised version to better illustrate its general applicability.

---

### Comment · Area_Chair_zPss · 2025-08-04
**Reminder: Author–Reviewer Discussion**

Dear reviewers,

A friendly reminder that the author–reviewer discussion period will close at August 6, 11:59 pm, AoE. Please engage with the authors’ questions and comments and update your Final Justification accordingly.

Thank you for your time and engagement.

Best regards,

AC

---

> ### Author Response · Authors · 2025-08-05
>
> **Dear Area Chair,**
>
> Thank you very much for your kind reminder and efforts in managing our submission.
>
> We have carefully addressed all reviewer comments in our rebuttal, including additional analyses, discussions, and new experimental results. We would be happy to provide further clarifications if needed before the discussion period concludes.
>
> Finally, we sincerely thank the reviewers for their time and constructive feedback.
>
> **Best regards,**
> The Authors

---

### Decision · Program_Chairs · 2025-09-17

**Decision:**

Accept (poster)

**Comment:**

This paper presents a new continual learning method named Continuous Subspace Optimization (CSO), which maintains and updates a subspace representation for each task using online PCA. The method aims to preserve past knowledge by minimizing interference with previously learned subspaces and by using regularization strategies that guide optimization toward underexplored parameter directions. Empirical results show improved forgetting and stability metrics across standard continual learning benchmarks. The paper is clearly written, the method is simple and modular, and the ablation studies are generally well executed.

Despite the overall positive reception, reviewers raised concerns about novelty and depth: subspace-based methods are not new, and the proposed update mechanism, while efficient, appears incremental. Furthermore, the theoretical motivation is limited, and the analysis of interference and orthogonality is mostly empirical. The rebuttal clarified the conceptual distinctions from prior work, and reviewers acknowledged the method's practicality and performance. Given the competitive results, strong presentation, and ease of implementation, I recommend acceptance, though not at the level of spotlight.